# Biomimetic Artificial Proton Channels

**DOI:** 10.3390/biom12101473

**Published:** 2022-10-13

**Authors:** Iuliana-Marilena Andrei, Mihail Barboiu

**Affiliations:** Adaptative Supramolecular Nanosystems Group, Institut Europeen des Membranes, University of Montpellier, ENSCM-CNRS, Place E. Bataillon CC047, 34095 Montpellier, France

**Keywords:** ion-channels, bilayer-membranes, proton transport, self-assembly, H-bonding

## Abstract

One of the most common biochemical processes is the proton transfer through the cell membranes, having significant physiological functions in living organisms. The proton translocation mechanism has been extensively studied; however, mechanistic details of this transport are still needed. During the last decades, the field of artificial proton channels has been in continuous growth, and understanding the phenomena of how confined water and channel components mediate proton dynamics is very important. Thus, proton transfer continues to be an active area of experimental and theoretical investigations, and acquiring insights into the proton transfer mechanism is important as this enlightenment will provide direct applications in several fields. In this review, we present an overview of the development of various artificial proton channels, focusing mostly on their design, self-assembly behavior, proton transport activity performed on bilayer membranes, and comparison with protein proton channels. In the end, we discuss their potential applications as well as future development and perspectives.

## 1. Introduction

Cells are the building blocks of any superior living organism, surrounded by their membrane, separating the intracellular space from the extracellular environment. From a functional point of view, cell membranes are complex self-regulating systems. One of the most important functions of the cell membrane is the regulation of metabolite transport for the cell. Due to the structural incompatibility of the hydrophilic protons and the hydrophobic lipid membrane components, they cannot diffuse through the cell membrane, and their transfer may be facilitated by proton channels or proton carriers [1]. Natural proton channels are protein voltage-gated pores that have proton selectivity over other ions, and their voltage dependence is similar to the voltage-gated channels for Na^+^, K^+^, and Ca^2+^ (the membrane permeability increases on depolarization). At physiological pH, the H^+^ ions are at low concentrations on both sides of the cell membrane, and their voltage-gated dependency is controlled by pH gradient [2,3]. The progress made until recently shows that, along with the membrane-bound water molecules, protons are translocating the pores via a Grotthuss-type mechanism [4,5,6].

While investigating the transfer mechanism of natural proton channels, intensive research efforts are made to design synthetic proton channels using artificial compounds. Inspired by biological proton channels, a big number of biomimetic artificial proton channels have been developed and studied, such as metal–organic frameworks (MOFs) [7], self-assembled pillar[5]arenes [8,9], or I-quartet supramolecular channels [10]. These artificial superstructures are constructed by employing numerous reversible non-covalent bonds, for example, H-bonding, electrostatic interactions, metal-coordination, or ion-π interactions. Supramolecular artificial proton channels designed as a biomimetic approach to replicate the self-assembly process of natural proteins in bilayer membranes in both structures and functions can present multiple advantages, such as the use of a simpler molecular components, stability towards the modification of physical constraints (temperature, pH, etc.), controllable transport properties, versatility and diversity due to their adaptive properties. These behaviors can help us clarify and understand the transport mechanism and the correlation between structure and function.

A biomimetic approach to the design of supramolecular proton channels has been defined as to strive in replicating the function, and not the structure, of certain natural functional units using artificial compounds. As a bioinspired approach, the aim is to take or to learn a few key elements from the known biological protein scaffolds and implement them in artificial systems. The synthetic strategy applied for the construction of functional pores for selective binding of water/proton superstructures is to self-assemble simpler molecular components through non-covalent interactions and in a controlled manner to ensure the directional translocating pathways. The aim of this constitutional synthesis is to simultaneously present high permeability for water and the ability to reject ions at a significant level. An optimal H-bonding of water clusters within the channel is needed for selectivity, while fewer interactions with the channel structure are desired to increase the water permeability.

For the construction of molecular components, it is of great importance to take into consideration three major factors: (a) the lipophilicity as this is determining the insertion of the supramolecular structure into the lipid bilayer to form the channel; (b) the overall length of the channel that needs to be long enough to span the phospholipid bilayer membranes thickness ~35 Å; (c) the water/proton recognition sites to ensure the selectivity of the transport. The interactions between the recognition sites and translocating waters/protons should not be too weak or too strong. It must be an optimal association constant for obtaining good transport selectivity [10,11,12,13,14].

In this review, the main focus is to present the state of the art for the development of biomimetic artificial proton channels via supramolecular self-assembly. The discussion will be divided into two parts: natural protein proton channels and artificial proton channels, focusing mostly on the recent discoveries and highlighting the emerging concepts used for the design of biomimetic artificial water/proton channels, the resulting self-assembled channel, and the structure–function relationship.

## 2. Natural Protein Proton Channels

In 1982, Thomas and Meech reported and proved for the first time the existence of voltage-gated proton channels. The experiments were performed in isolated suboesophageal neurones of snails, Helix aspersa, by measuring the hydrogen ion currents and intracellular pH in voltage-clamped neurons. The cell body was voltage-clamped by four microelectrodes, in which the first was to measure the pH, and HCl was microinjected through a fourth microelectrode. The intracellular pH, as well as the membrane currents, were monitored after the microinjection of the HCl, and it resulted that the pH recovered within 10 min. The condition involved the exchange of external Na^+^/HCO_3_^−^ with the internal H^+^/Cl^−^. The exchange of the antiporters was inhibited by changing the external medium with a bicarbonate-free solution. After the HCl injection, the intracellular pH recovered faster and by a different mechanism when the membrane potential was depolarized. Moreover, the experiments showed that pH recovery can be inhibited by heavy metals present in the extracellular medium [2].

The studies performed in molluscan nerve cells proved that the membrane permeability to the H^+^ transport increases with membrane depolarization. Holding the potential at different positive values under a voltage clamp changes the pH in the cell’s cytoplasm. Thomas and Meech reported more series of experiments in which they used intracellular pH-sensitive microelectrodes to show the effect of the membrane potential. They showed that the addition of Cd^2+^ in the bathing medium could reversibly block the H^+^ pathway. Following the measurements recorded previously by using four microelectrodes in order to trace the membrane potential (Em), clamp current, injection current, and pH, in which the HCl was microinjected through one of the microelectrodes at two different holding potentials, −50 and +15 mV [2], in these experiments the membrane potential was measured with a reference liquid ion exchanger microelectrode. The microelectrode was checked at the end of the assay with a 3 M KCl-filled micropipette. The potential difference between the calibrated pH-sensitive electrode and the KCl-filled micropipette gave the pH of the cell cytoplasm (Figure 1) [3].

During the last decades, part of the research attention on proton transfer reactions was focused on the voltage-gated proton channel of Hv1 as gaining more knowledge in this domain will have a direct impact on threatening several diseases. This type of protein channel is found in diverse species, from unicellular marine life to humans. However, mammalian phagocytes were originally proposed to be participating in the NADPH oxidase respiration, this enzyme being an inside source of O_2_ [15].

The actual process happens in mitochondria which are known to be the “power plants” of the cells. The electrons generated by NADH (reduced nicotinamide adenine dinucleotide) are subjected to a series of respiratory processes catalyzed by enzyme complexes which are located in the inner mitochondrial membrane. The energy resulting from this electron transfer is used to pump protons across the membrane and is then used by ATP synthase to make ATP (adenosine triphosphate) from ADP (adenosine 5′-diphosphate) and phosphate. ATP is used to fuel motion or transport anywhere in the cells of living organisms (Figure 2). Protons are essential in bioenergetics, and the proton transfer reactions in biochemistry continues to be an area of active interest [1,16].

This membrane protein channel allows the flow of H^+^ outside of cells resulting in changing the pH on both sides of the membrane. The electrochemical potential difference across the cell membrane forces the voltage-gated proton channels to respond to it, resulting in selective proton transport by opening in response to the depolarization of the membrane and closing with hyperpolarization (Figure 3) [17]. This property of Hv1 channels helps to maintain the intracellular pH neutral and regulate it in the cytoplasm, making it alkaline [18].

Recent studies showed that the Hv1 channel in a marine animal species, Ciona intestinalis VSP, has a voltage domain sensor. The carboxyl-terminal (C terminus) domain of the channels is known to be responsible for the dimeric architecture, and each subunit contains its own pore, but the role and structure of the C-terminal of Hv1 are known. Apparently, studies conducted on HeLa cells, cancerous cellular lines deducted from metastases, illustrated that the C terminus is essential for Hv1 localization. Two monomers form the dimer through parallel α-helical coiled-coil interactions. Results implicate that the protons’ permeability might be regulated by the C terminus through its structural changes, which are inwards pH dependent (Figure 4) [19,20,21].

In 1992, Pinto et al. showed that the M2 protein of the Influenza A virus has ion channels activity [22]. Later discoveries of the M2 protein of Influenza A and B viruses reveal ion channel activity that conducts protons across the membrane, an important fact for the viral life cycle [23,24]. The two predominant types of Influenza viruses that infect humans remain a fundamental challenge in the research field of biochemistry and chemistry since the discovery of the M2 protein as an ion channel [24]. The M2 protein is a pH-gated channel and becomes activated after the Influenza virus enters cells via endosomes. The low endosomal pH activates the M2 channel and allows the proton transport into the viral interior, thereby triggering the disassociation of the viral RNA from the protein matrix and the fusion of the viral and endosomal membranes, required interaction for infectivity [23,24,25,26,27]. In later stages of virus replication, M2 maintains a high pH of the trans-Golgi network membrane, this way it prevents the premature conformational rearrangements of hemagglutinin in viruses to a low pH form [24,26,27].

M2 is one of the smallest ion channel proteins, containing a single transmembrane helix, which tetramerizes into four tightly packed transmembrane helices to form the conducting pore [26,27]. This is a single-pass membrane protein containing 97 amino-acid residues with an amino terminus toward the outside and the carboxy terminus toward the inside of the virion. His37 is located near the center of the transmembrane helix, acting as the pH-sensor of the channels, and Trp41, located next to the His 37 towards the C-terminal, is the gate for proton transfer. The acid activation mechanism occurs when the pH decreases near 6.0, His37 serving as a sensor activates the channel, and Trp41 side chains open for the proton flow. As the pH level decreases to ~6.0, the protonation conditions of His37 reach high pH conditions where Trp41 side chains close the C terminus pore below His37, this way shaping the gate which blocks the proton transfer through the channels [26,27,28]. The transport activity of the M2 channel has been studied extensively, and the suggested mechanism for the selective proton conduction is dependent on the outside low pH, as well as on the interactions with the water-filled pores. Protons first bind to the histidine residues (His37), which act like a selectivity filter, and the water wires formed across the conducting pathway. The hydrodynamics of the water through the channel will strictly depend on the channel–water and water–water interactions (Figure 5) [26,27].

Another related example of a natural protein proton channel is the urel valve, which functions as a proton-gated urea channel controlling the cytoplasmic urease, an enzyme that is important for gastric survival and colonization of Helicobacter pylori. The infection with the gram-negative pathogen Helicobacter pylori affects almost half of the world’s population, causing gastritis, gastric ulcers, and a high incidence of stomach cancer. This unique ability to colonize and survive in such acidic conditions inside the stomach is strictly dependent on the urea transport through the proton-gated channel, HpUrel. This is an inner membrane protein containing six transmembrane segments with acid-dependent properties that activate the urea entry into the cytoplasm at low pH and maintains a proton gradient across the bacterial membrane. The urea transport through HpUrel is specific, passive, non-saturable, non-electrogenic, and temperature-independent [29,30].

The proton-gated inner membrane urea channel is closed at neutral pH and opens at low pH to permit the access of urea to reach the cytoplasmic urease. The enzyme produces NH_3_ and CO_2_ to neutralize the entering protons and thus maintaining the periplasmic pH around 6.1. The six transmembrane segments of the channel assemble in a hexameric ring, with the center filled with asymmetric lipid bilayers. The formed pore is lined by amino-acid side chains that most probably identify and select the solutes (Figure 6). Unfortunately, even with the latest discoveries on channel architecture, the transport mechanism and selectivity for urea remain unknown. HpUrel channels can also conduct water the same way as urea, in a pH-dependent manner and sharing a common transport pathway [31,32].

## 3. Artificial Proton Channels

The research field of synthetic proton channels is gaining more and more attention among the biochemistry and chemistry research groups. The general approach is bioinspired, starting from the natural protein proton channels, and mostly modified peptides and proteins are used for the development and understanding of the mechanism of proton transport. One other common biomimetic approach is related to the synthesis and characterization of novel molecules, easier to produce, more stable, and cost-effective, such as active chemical compounds. When discussing artificial proton channels, there are a few considerations to keep in mind. One criterion would be the structure; simple molecules are preferred to more complex systems, as a complex system would be more difficult to characterize by conventional methods used in proton and water transport.

Another important aspect would be the compatibility with the lipid bilayer, which comes as a consequence of the characterization methods. Extensive research was dedicated to developing such simple systems but with a big impact on the attempts to elucidate the proton transfer mechanism.

Theodore von Grotthuss proposed in 1806 a theory that water can decompose into charged particles that can move quickly along the water wires. [33] Almost two hundred years later, the Grotthuss mechanism, described as fast proton hopping along the hydrogen-bonded water wires immediately followed by the reorientation of water molecules, is still accepted as an important mechanism that gives a diffusion coefficient with an order of magnitude greater than any other hydrated ion for protons in aqueous solutions [34].

Proton transport is of great importance in biological applications, and the implementation of highly efficient proton transport in biomimetic artificial channels has not been very successful, as the research in this domain is still ongoing. Hydrated Nafion proton exchange membranes are the most attractive polymer electrolyte developed so far, with the most efficient proton conductivity when soaked in water, yet their intrinsic proton conductivity is still lower than that of Gramicidin A. Molecular dynamic simulations performed on perfluorinated sulfonic acid polymers such as Nafion membranes showed high proton conductivity dependent on the water content, porosity, distribution of protons, and the ratio of diffusion coefficients [35,36].

Important studies have been carried out based on the morphology and its effect on water swelling and proton conductivity with Nafion (Figure 7). The sulfonic acid groups in Nafion influence the acidity and the hydrophobicity and flexibility of backbones. The Nafion membranes are reported to successfully form proton conductive channels as a result of the polarity difference between the hydrophobic backbones and the hydrophilic side chains. The lack of the side chains’ flexibility results in poor proton transport as it inhibits the aggregation of ionic sites. However, there are some disadvantages to using Nafion commercial membranes, such as the cost and the dehydration, which leads to the loss of conductivity at higher temperatures [37].

Nafion membranes are a matter of continuing studies to improve their transport properties. Sulfonated pillar[5]arene/Nafion composite membranes were investigated and tested in direct methanol fuel cells for proton selectivity and conductivity; these novel materials exhibited properties far superior to Nafion. As direct methanol fuel cells are considered to be promising power-generating systems with valuable applications such as portable electronic devices, the proton exchange membrane is an important component of these systems in preventing the methanol crossover and allowing the proton transport from anode to cathode. A promising approach has been the incorporation of pillar[5]arenes into Nafion membranes. Pillar[5]arenes are known to form artificial water channels in lipid bilayers, and along the water wires formed across their channel, protons migrate via the Grotthuss-type mechanism with complete rejection of other ions [6,34]. Pillar[5]arenes were included as filler in Nafion membranes, performing excellent water swelling at room temperature and good proton conductivity, which increases with the increasing of the pillar[5]arene concentration [38].

Proton exchange fuel cell membranes represent a promising clean technology for high-efficiency power generation. Their main application is as the power source for an electric vehicle having zero CO_2_ emission. The Proton exchange membrane is an important component, and over the past decade, attention has been focused on developing new hybrid materials for optimal proton transport. New hybrid materials based on carbon nanotubes rooted in smectite clays were designed and studied to be introduced in a Nafion membrane [39]. Known for their wide applications in nanotechnologies, carbon nanotubes have attracted more attention for applications in nanofluidic devices for translocation and separation [40,41]. The nanocomposite materials based on clay-carbon nanotube additives are expected to show a significant improvement in water retention at high temperatures and thus will influence the proton conductivity. The novel Nafion composite membranes were obtained by the solution-precipitation method and then were structurally and morphologically characterized, proving that the clay-carbon nanotubes were successfully dispersed and the membrane was homogenous. The carbon nanotubes obtained on the clay platelets were oxidized and functionalized with alkyl-sulfonic groups to ensure an efficient proton transport of the nanocomposites. The hybrid materials present a good proton conductivity (7 × 10^−2^ S cm^−1^), a very high value compared to the other Nafion-based composites. The network formation of these composites favors proton transport by a Grotthuss-type mechanism [39].

Carbon nanotubes have been easily proven to self-insert into biological and synthetic lipid bilayer membranes having high permeability for water and ions without any selectivity. It was observed that faster water and proton transport is mostly occurring when the carbon nanotubes’ diameter is in the range of 0.8 nm. In this case, the water transport exceeds that of biological aquaporins-AQPs water channels. Water transport has a big influence on the proton transport rates as it is directed by the organization of the water molecules into a water wire, reaching values of one magnitude higher than bulk water. Water clusters are broken down upon entering the carbon nanotubes, and it suffers a transition from bulk-phase water to single-file chain-organized water [42].

Noy et al. reported the important role that molecular confinement of water wires in narrow hydrophobic pores of carbon nanotubes has in enhancing the rates of proton transport via the Grotthuss-type mechanism [34]. They confirmed that the measured values of proton conductance within a 0.8 nm cavity exceed that of the Gramicidin A channel and of Nafion. Multiple molecular dynamic (MD) simulations have been carried out on the proton conduction along the water wires to characterize the proton transfer events that appear through carbon nanotube pores. The MD simulations also revealed the preference of water-filled carbon nanotubes to adopt a well-aligned water wire resulting in excess proton conduction along with this water wire [43,44,45].

Biomimetic artificial proton molecular or supramolecular self-assembled channels attracted substantial interest, and their transfer mechanism might be of great importance in elucidating the way natural protein proton channels function.

Rigid-rod polyols represent a class of nonpeptide ion channel models which have been studied in the hope of contributing to the understanding of the way protein channels function at the molecular level and developing new pharmaceuticals useful for treating certain diseases. This class of compounds studied for their ion and proton transport properties were designed to act as unimolecular tunnels (Figure 8) [46].

The synthesized octamer having a length of 34 Å nearly matches the thickness of the bilayer membranes (35 Å). Its ion transport activity was evaluated with small unilamellar vesicles (SUV) composed of EYPC containing 8-hydroxypyrene-1,3,6-trisulfonic acid (HPTS). It was observed that the octamer acts as an H^+^/K^+^ exchanger with significant H^+^ selectivity over K^+^ cations. Further investigations were conducted and it was found that the octameric polyol reveals its selectivity over the transport of other cations (Rb^+^ > Cs^+^ > K^+^ > Na^+^ ~ Li^+^ > Mg^2+^ > Ca^2+^). The rigid-rod octameric polyol acts as a selective unimolecular proton channel in SUVs and represents the first described nonpeptide channel model of the hydrogen-bonded water chain mimic mechanism [46].

Still far from being fully explored, functionalized rigid-rod polyols were further investigated as nonpeptide artificial channels for their ion and proton transport ability. New molecules in which lipophilicity was increased for better insertion into the lipid bilayer were designed and tested for their transport selectivity (Figure 9) [47]. Their successful incorporation showed promising results for ion transport [48,49].

Another approach to designing new synthetic channels for proton transport is based on metal–organic frameworks (MOFs). Recent studies presented water-stable sub-1-nm MOFs promoted for proton conduction. These novel artificial channels present permanent channels porosity, tunable pore size, and adjustable surface properties (Figure 10) [50]. The sulfonated sub-1-nm MOF channels are able to transport water and protons fast and selective, but in comparison to other types of artificial water and proton channels, these functional MOFs are able to conduct unidirectional due to their uniform pore structures. The water molecules pass through the MOF channels as water wires forming hydrogen-bonded chains for fast proton transport via the Grotthuss mechanism [51].

Further investigations in the development of tubular structures to allow selective transport of protons across the membrane led to a new kind of supramolecular structure, pillar[5]arenes (PAs), synthesized and characterized for the first time by Ogoshi et al. [52]. PAs are reported to induce the formation of water wires, which could mediate the H^+^ transport across the lipid bilayer. Hou et al. observed for the first time that PA and its derivatives form tubular structures possess tunable size cavities that could be suitable platforms for constructing artificial channels (Figure 11) [8,9]. Vesicular transport experiments and Molecular Dynamic simulations revealed selective water transport with total ion rejection due to an inner pore size of ~5 Å. The channels exhibit water permeability at a very low channel-to-lipid ratio and selective proton conductance [8,9,53].

The development of artificial aquaporins that can work in cell membranes with high water permeability and ion selectivity by using pillar[5]arene scaffolds was widely studied by Hou et al. The pillar[5]arene macrocycle has been used to create tubular structures that can encapsulate single water wires. There was reported the construction of three different channels by attaching side chains with different charges of the amino groups at both entrances onto a pillar[5]arene scaffold. The first molecule contains positively charged amino groups, the second one is a zwitterionic channel, and the third design is a negatively charged channel (Figure 12). These compounds were incorporated into bilayers and investigated for their water transport efficiency. As great water transport results were achieved with the first type of designed molecule, the channels were further investigated in cell membranes and indicated ion-exclusion behaviors. MD simulations monitored the orientation of water molecules through the assembled channels and proved that the narrow region of the channels prevented the formation of proton wires along the single water wire file, which represented total proton rejection [54].

A recent publication by Barboiu and Ogoshi groups reported the synthesis of rim-differentiated pillar[5]arenes and their assembly into dimers via hydrogen bonding produced an expanded length-controlled tubular superstructure with high water permeability (one order of magnitude lower than that of natural aquaporin) and total ion rejection, but also with total proton rejection (Figure 13) [55,56]. In comparison to the PAs reported by Hou et al. [53], these rim-differentiated pillar[5]arenes had an unexpected behavior; protons do not cross the lipid bilayer through the assembled channels [56].

Further studies were carried out on pillar[n]arenes by Hou et al. as rigid frameworks that allow selective transport of protons. It was reported that new pillar[n]arene (*n* = 5 and 6) derivatives can successfully transport protons, water, amino acids, and K^+^ in a selective manner as unimolecular channels. Ester-attached PA[5] derivative was incorporated into the planar lipid bilayer as proton channels. The proton transport across the lipid bilayer membranes was monitored by the patch-clamp experiments carried out with hydrochloric acid solution (HCl/H_2_O). Dimeric PA[5] units with an aliphatic linker were synthesized and tested for their proton activity to ensure a better comparison (Figure 14). As expected, the proton exchange rates were different, and the PA derivative with four carbons aliphatic linker (2c) had the highest proton activity (Figure 15) [8,9].

Cucurbit[n]uril derivatives (*n* = 5 and 6) are promising structures that can transport protons and alkali ions across lipid membranes with ion selectivity. Cucurbit[6]uril is a macrocyclic molecule comprising six glycoluril units with a pore (~5.5 Å diameter) that is accessible through two identical carbonyl-fringed portals (~4.9 Å diameter). Cucurbit[5]uril has five glycoluril units and the smallest cavity (~4.4 Å diameter) and portals (~2.4 Å diameter) (Figure 16) [57].

These molecules are attractive not only as synthetic channels but also as building blocks for supramolecular assemblies. For the proton assays across lipid membranes, it was used egg yolk phosphstidylcholine (EYPC), cholesterol, and dicetyl phosphate vesicles. The proton transfer through the vesicles’ membrane was assessed with fluorimetric HPTS assays. The change in the fluorescence intensity certified that cucurbit[n]uril (*n* = 5 and 6) channels can transport protons and ions across the lipid bilayer with remarkable selectivity for ions (especially for K^+^) (Figure 17) [57].

Artificial foldamers might be an optimal choice for artificial proton channels, as they possess well-defined backbones and adjustable cavities. Recently, it was reported a novel biomimetic artificial proton channel based on a quinolone-derived helix with a cavity diameter of 1 Å can selectively filter ions and even water, giving access through the channel only for proton flux. The synthetic nanopore contains NH groups and serves as a pathway for high proton translocation without the permeation of water molecules. HPTS fluorescence assays were conducted in order to examine the proton transport.

Egg yolk phosphatidylcholine (EYPC) large unilamellar vesicles (LUV, pH 7.0) entrapping HPTS were prepared for the conducting fluorescence experiments. The intensity variation was continuously monitored as the permeation of H+ into the vesicles was performed through the helix-based channels. The results obtained indicated better proton transport properties from pH 6.4 to 7.0 than from pH 7.0 to 7.6 across the membranes. The single-channel electrophysiological experiments showed a high selectivity over Na^+^, K^+^, Cl^−,^ and good proton transport (107 H^+^/s/channel) (Figure 18) [58].

Among the natural channels of low specificity, Gramicidin A is the simplest known channel and is especially relevant for the cation and proton conductance properties. This natural channel has been widely studied as it has been considered to be the poster child for the Grotthuss mechanism, where protons are envisioned to diffuse along the water-filled pores [59,60]. Starting from a bioinspired approach, Barboiu et al. proposed an artificial primitive mimic of Gramicidin A. The artificial system obtained mimics the functions of the natural channel and is constructed by using a simple synthetic triazole (‘T-channel’) [61,62]. The bola-amphiphile-triazole compound (TCT) has the capacity to self-assemble via hydrogen bonding into stable channels in lipid bilayer membranes. The self-assembled pores present internal diastereoisomeric chiral hypersurfaces on which water wires arrays can form having a dipolar orientation. This biomimetic artificial system presents ion, proton, and water conductance properties in lipid bilayers (Figure 19) [61].

The proton transport experiments were performed on HPTS-loaded vesicles, and the activity was measured by continuously monitoring the fluorescence intensity variation. The transport was assessed by pH gradient assays. The change in the internal pH under osmotic conditions might support the assumption that the T-channels produce a water influx and induce an efflux of protons outwards the vesicular solution. The vesicles grow in size due to the osmotic swelling, causing the ionic balance between the lipids to modify, which might be compensated by the absorption of ion pairs. Proton-chloride (H^+^/Cl^−^) symport (unidirectional transport of both species) could occur to directly increase the internal pH. It was observed that the proton transport has low conductance under non-osmotic conditions compared to the osmotic conditions; the transport may occur according to the Grothuss mechanism [50] along oriented water wires similar to that in Gramicidin A channel [61].

An extended study was conducted on the self-assembly of the triazole amphiphiles for their ion transport. A different approach was to construct novel anion channels with precise recognition in the selectivity filters. A selective anion recognition has to be combined with fast anion translocation along the directional pathways in channels. A series of protonated amino-triazole amphiphiles were synthesized and characterized by Barboiu et al. to form self-assembled channels of stacked triazole quartets (Figure 20) [63].

The artificial channel’s template contains hydrogen-bonded anions that interact through anion-π interactions with triazoles of the vicinal quartets to self-orient translocation along the anion-selective cavities. This combination of classical hydrogen bonding/ion pairing with anion-π interactions creates channels selective for anion binding with pore openings between 3 to 4 Å wide and 9 to 10 Å length. The ion transport activity was evaluated by HPTS assays into EYPC liposomes. A series of experiments were performed by using an H^+^ selective carrier (FCCP) or a K^+^-selective carrier (valinomycin). There was no significant difference in the transport activity, confirming that these artificial triazole channels have no cation selectivity (Figure 21) [63].

Over the past decade, a variety of artificial water channels based on unimolecular and supramolecular systems have been reported to achieve rapid water transport across bilayer membranes; some of them lack the stability and compatibility of AQPs outside the cellular environment, limiting this way their applications for water desalination technologies. Barboiu et al. reported for the first time in 2011 that the imidazole-quartet (I-quartet) channels as excellent candidates for water/proton translocation, whereas the term “Artificial water channels” was first proposed independently in 2012 by Barboiu [64] and Hou [53,54]. Self-assembled ureido imidazole derivatives were used as molecular scaffolds to construct the I-quartets stabilized by inner-oriented dipolar water wires [65]. The I-quartets are stable channels in bilayer membranes and can transport the water and protons selectively with total ion rejection. They are able to transport ~1.5 × 10^6^ water molecules/s/channel, which is within two orders of magnitude of aquaporin proteins. The proton conductance of these channels is also quite high ~5 H^+^/s/channel which is approximately half of the proton’s value that natural protein M2 channels of Influenza A virus at neutral pH can transport. With an inner pore of 2.6 Å, the I-quartets can effectively transport the protons only when an osmotic pressure is applied, creating this way water transport gradients (Figure 22). The channel’s inner water wires allow the synergic antiport proton translocation through the lipid bilayer. Overall, these self-assembled I-quartet channels, obtained using simple chemistry reactions, can be assembled into artificial water/proton channels and remain stable in the lipid bilayers [65].

Later in 2021, Barboiu et al. reported octyl-ureido-polyol artificial water channels capable of self-assembling into hydrophilic hydroxyl channels with absolute water selectivity (Figure 23). Interestingly as it may be, these OH channels are capable of accommodating the single water wires to selectively transport the water across the lipid membranes with total ion and also proton rejection. Even though these OH channels have a high-water permeability as I-quartets and their structure are approximatively similar, the M^+^/H^+^ antiport-opposite transport of the species-conductance states was not observed for any of the tested concentrations [66,67,68].

On the same line, diol peptide isomers attracted interest for their properties of transporting water in a selective manner, like natural AQPs with total ion rejection. Six compounds were evaluated to transport water at high rates (5.05 × 10^8^ water molecules/s/channel), which is around one order of magnitude lower than that of AQPs. The peptide-diol dimers aggregate in a stacked cylindrical channel that forms an adequate cavity for water recognition and salt exclusion (Figure 24). The water and ion transport of these structures were evaluated into lipid bilayer membranes, and the obtained results showed high water permeability and no cation, anion, or proton transport (Figure 25) [69].

Active artificial channels involved in water and ion transport may also be constructed from simpler amphiphilic compounds presenting an aromatic chiral hydrophobic group acting as the component that interacts with the lipid bilayer membrane and a heterocyclic imidazole or amino-triazole for water/ion interaction within channel type superstructures (Figure 26). Minor structural modifications were made to the aromatic rings for a strategic characterization of their potential ion and proton transport. The compounds’ proton transport was assessed by membrane polarization via valinomycin, known to generate strong K^+^/H^+^ antiport. The overall proton activity of the channels presented weak proton transport, except for compounds four and five, which displayed the best activity transport towards H^+^ [70].

Transmembrane proton-coupled anion transport has attracted interest over the past decades, encouraging the development of synthetic chloride (Cl^−^) transporters. Therefore, the need to design novel artificial channels with cotransport properties has led to new synthetic systems capable of H^+^/Cl^−^ symport transport

Amino-imidazole amphiphiles that can self-assemble via hydrogen bonding can form stable artificial Cl^−^ pH-responsive channels within lipid bilayers (Figure 27). These synthetic channels, regulated by pH, can generate a Cl^−^/H^+^ symport at acidic pH or Cl^−^/OH^−^ antiport at higher pH. The voltage/pH-regulated systems can represent an alternative to the ion-pumping along the artificial ion channels. Similar to the function of the intracellular chloride channel proteins, these artificial voltage-responsive channels can become activated at high basic pH. The systems can be considered to be pH/voltage-triggered biomimetic Cl- channels as they show dual responsive properties with ion transport activated under low and high pH and the application of high membrane voltage potential. Under weakly acidic conditions, there is a Cl^−^/H^+^ symport conductance along a partly protonated channel mainly driven by hydrogen bonding interactions [71].

A series of bis(amino)-imidazole compounds were studied for their ion transport properties, confirming by HPTS and lucigenin assays that there is an H^+^/Cl^−^ symport process dependent on the applied pH gradient conditions. Using the same pH in the feed and receiving phases, there is also anion antiport transport. In the case of ion selectivity, two ways of a possible transport mechanism were approached; either H^+^/A^−^ symport or Cl^−^/A^−^ antiport. FCCP and valinomycin were used under pH gradient conditions to assess clear results (Figure 28). FCCP allows the efflux of H^+^ under an applied pH gradient, and the Cl^−^/A^−^ antiport transport rate should increase. However, there was no difference seen in the absence or presence of FCCP, which concludes that the proton transport most probably occurs via the symport H^+^/A^−^ [72].

Phenyl-thiosemicarbazones (PTSCs) might be another class of compounds used for proton-coupled anion transport with pH-dependent behavior. The study was conducted to achieve a better understanding of the behavior of pH-dependent anion transporters. The H^+^/Cl^−^ symport transport was performed on large unilamellar POPC vesicles loaded with KCl at different pH, and it was observed that H^+^/Cl^−^ symport facilitated by the anion transporters is required to generate KCl efflux [73].

Random heteropolymers (RHP) can be easily synthesized, and their chains contain two or more monomers arranged in random sequences. Recent discoveries showed that random heteropolymers can mimic proteins, and they are also able to perform certain natural functions such as providing support for protein folding, stabilizing proteins in nonpolar environments, helping proteins for better insertion into bilayer membranes, and forming artificial channels for proton transport in bilayer membranes [74].

In addition to all the systems presented above, heteropolymers might also be an interesting alternative for artificial proton channels, forming structured channels that can mimic membrane proteins and exhibit selective proton transport across lipid bilayers. Four monomer-based random heteropolymers presented selective and rapid proton transport. The statistical control over the monomers’ distribution leads to high segmental heterogeneity in hydrophobicity, thus facilitating the insertion into lipid bilayers. The proton transport was evaluated using liposome-based HPTS fluorescence assays with valinomycin as a K^+^ carrier. The segmental heterogeneity formed by the four distinct monomers appears to be the key design feature for transporting protons in a selective manner [75].

Further experiments conducted by using random heteropolymers showed that they were capable of interacting with proteins and mediating their interactions with the environment. RHP stabilized the proteins into DOPC liposomes, and to verify if the protein folded to retain its transport function in a lipid environment, HPTS-based proton transport assays were performed. By adding valinomycin to the outer solution containing the dipeptide Ala-Ala, proton transport was detected. Thus, random heteropolymers are capable of supporting the protein folding and helping the insertion of the protein into lipid membranes without outcompeting the protein–lipid interactions or compromising the protein transport functions [76].

## 4. Conclusions

Proton transfer is crucial in numerous biological and chemical processes, and its transport is facilitated by membrane proteins. Many details on the proton transfer mechanism are still unknown as many chemistry and biochemistry research groups are attempting to enlighten the way. When discussing natural protein proton channels, of central importance is the knowledge of how proton transfer reactions can be used in pharmacology for threatening certain diseases, such as preventing brain damage from ischemic stroke [77,78], cancer [79] by inhibiting the voltage-gated proton channel Hv1, chronic gastritis, duodenal ulcers and, stomach cancer [80,81] by disabling the Urel channel of Helicobacter pylori which transports urea and maintains the proton gradient across the bacterial plasma, and the list can further continue.

Moreover, taking into account the disadvantages of working with natural protein proton channels, an alternative to accomplishing a better understanding of the proton transport would be the design and development of artificial proton channels mimicking the functions of natural proteins. Nature already designed the functional groups, and by simple chemistry, humans can adapt them to construct synthetic channels with high selectivity and transport efficiency close to natural ones. Natural proteins may serve to tailor the functions of the synthetic systems and engineer optimized supramolecular assembly.

The non-protein artificial proton channels discovered for the past decades have shown promising results on efficient transport, like in natural systems, which is usually a one-way path to create synthetic systems at the level of natural functions or even beyond natural functions. Since they are easier to control and develop and they have better stability than natural channels, supramolecular dynamic channels can be characterized on a molecular level or supramolecular scale, might have better solubility, and are budget-friendly. The field of artificial proton channels is still underdeveloped, and many contributions are still needed. The main goal is to obtain the same efficiency and selectivity of proteins.

Developing novel synthetic artificial systems implicated in proton transfer processes might give a better perspective on the transport mechanism of protons through the natural protein channels. The use of simple molecules to create artificial channels can give an insight into the preferred structures that can host a selective and faster proton flux across the channels; thus, it can be associated with the structural parts of the natural proteins that generate selective proton transport.

In this paper, we described a diversity non-exhaustive artificial channel that can translocate water and protons but also molecules that were designed to have only proton transport. The field is still developing as this attracted the attention of multiple research groups in innovating and creating synthetic artificial systems. These systems could improve the quality of life by solving certain problems with respect to related proton transfer processes applied to medicine, energy, and environmental sciences.

## Figures and Tables

**Figure 1 biomolecules-12-01473-f001:**
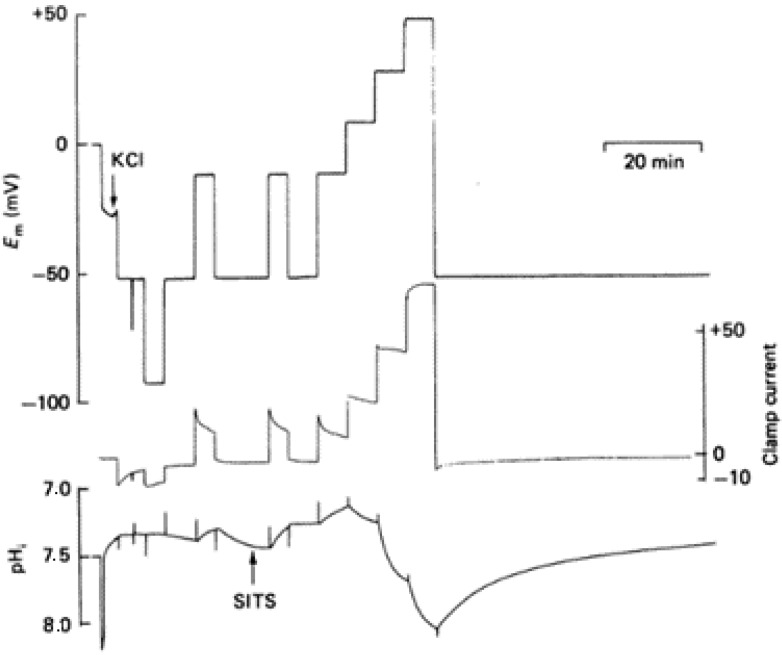
The experimental results of the voltage-clamped snail neurons. The measurements show the effect of different membrane potentials on pH. The cell was penetrated with the pH-sensitive electrode and the KCl-filled micropipette. When −52 mV were applied, the intracellular pH remained steady (~7.35), and a small change appeared when the holding potential reached −92 mV. After the membrane depolarization to −12 mV, the internal environment of the cell became acidic (~0.08). The pH recovered to ~7.4 after the repolarization of the cell to −52 mV. Throughout the experiment, a bicarbonate-free saline solution (pH = 7.5) was used, containing 10 μM 4-acetamido-4′-isothiocyanostilbene-2,2′-disulphonic acid (SITS) from the point indicated. The depolarization to −12 mV in the presence of SITS caused the pH to decrease to 0.2, and after repolarization, no recovery was recorded. Further depolarization produced further acidification within the cytoplasm. Adapted with permission from Reference [3] Copyright © 1987, The Physiological Society publishing.

**Figure 2 biomolecules-12-01473-f002:**
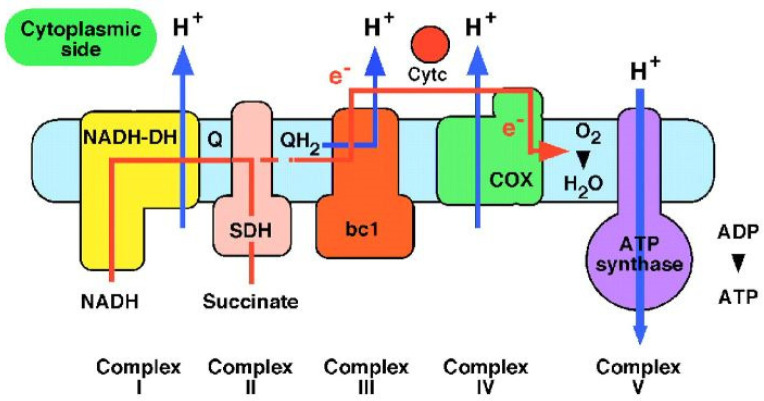
Regulation of ATP-producing pathway by oxidative phosphorylation in mitochondria. The pathway is accelerated when the use of ATP and the formation of ADP, AMP increases. Adapted with permission from Reference [16] Copyright © 1999, AAAS publishing.

**Figure 3 biomolecules-12-01473-f003:**
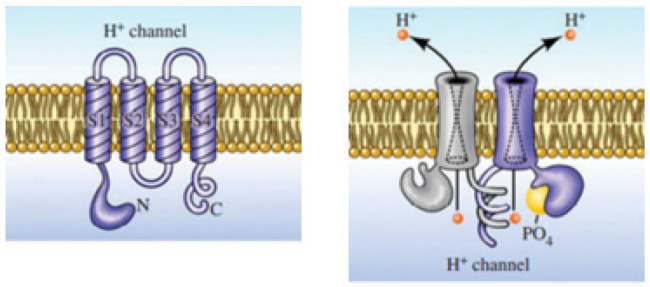
Architectural structure of the voltage-gated Hv1 channel. The top row represents the monomeric subunits of the channel, and the bottom row contains the complete protein. The Hv1 channel resembles four segments with no explicit pore domain. The four monomeric segments form a dimer through parallel α-helical coiled-coil interactions. Adapted with permission from Reference [17] Copyright © 2018, Royal Society of Chemistry publishing.

**Figure 4 biomolecules-12-01473-f004:**
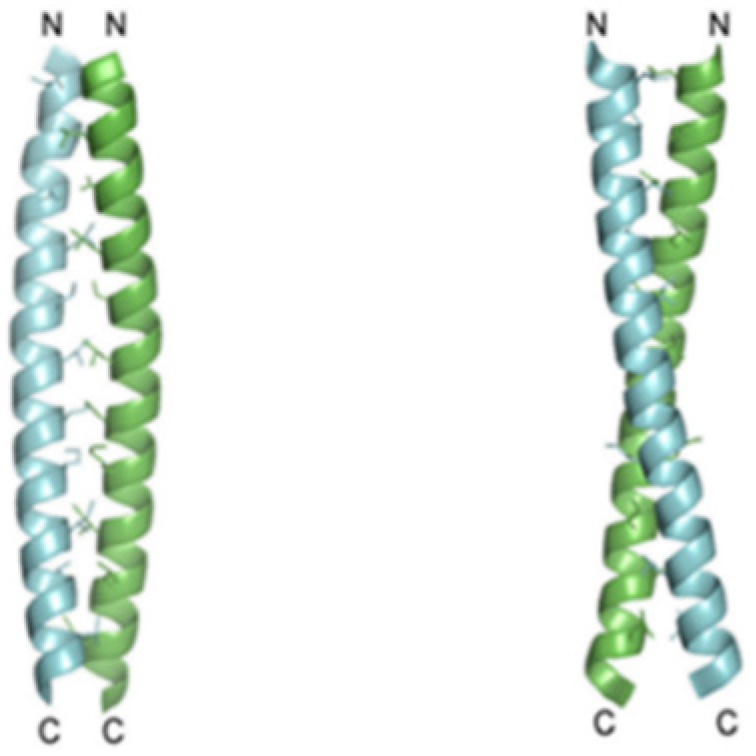
The structure of the C-terminal of Hv1. The parallel-oriented dimer forms a classic coiled-coil architecture. C terminus and N terminus are indicated. Adapted with permission from Reference [20] Copyright © 2010, The American Society for Biochemistry and Molecular biology publishing.

**Figure 5 biomolecules-12-01473-f005:**
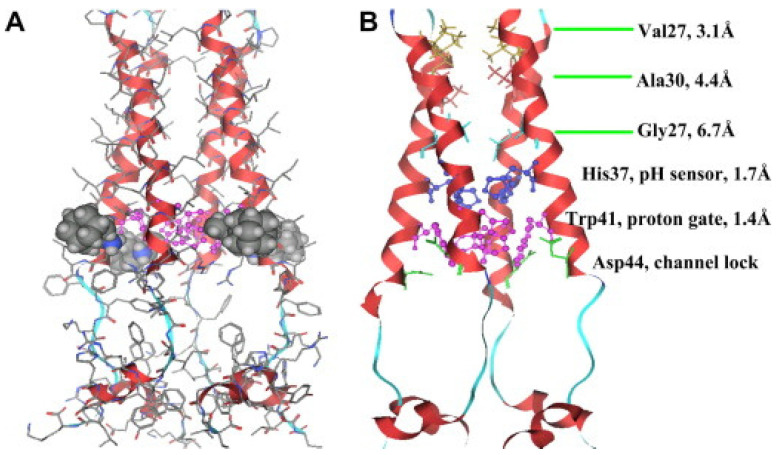
The structure of the M2 channel of Influenza A virus and water accessibility. (**A**) Tetrameric structuration of the M2 channel in complex with rimantadine determined by NMR. The four tightly packed helices form a left-handed twisting bundle forming a narrow pore in close state. (**B**) The pore surface of M2. The wide regions allow the flow of water and protons, but the narrow regions do not allow ions to pass through the channel. Adapted with permission from Reference [27] Copyright © 2008, Elsevier publishing.

**Figure 6 biomolecules-12-01473-f006:**
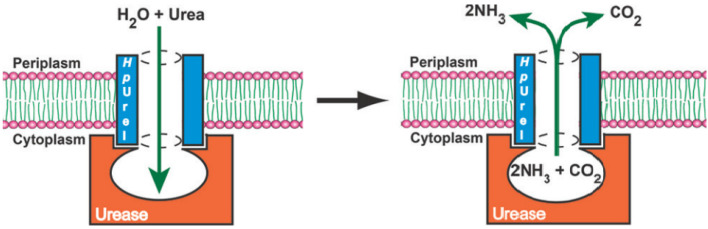
The structure of the urel protein channel and the conduction of urea to the cytoplasmic urease. The hexameric ring formed by six transmembrane segments arranged around a central channel filled with lipids disposed in asymmetric bilayers. Adapted with permission from Reference [32] Copyright © 2011, American Chemical Society publishing.

**Figure 7 biomolecules-12-01473-f007:**
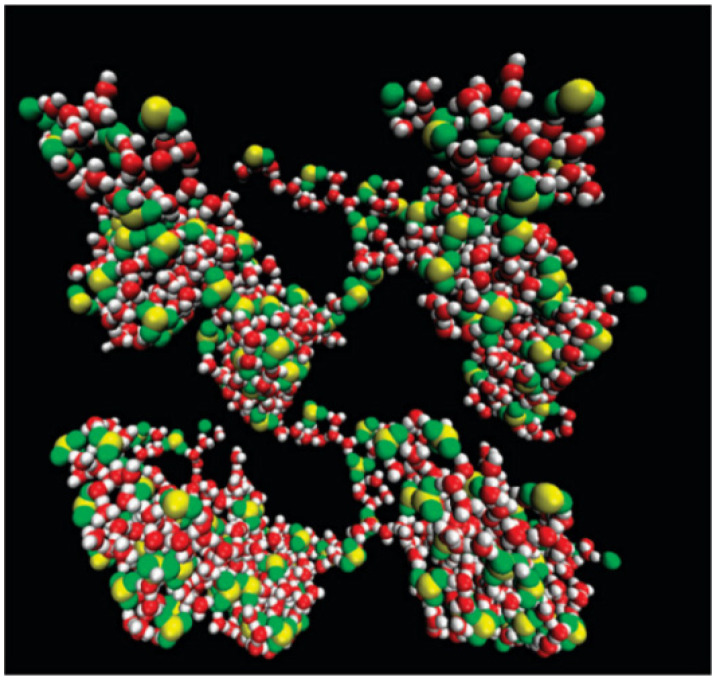
Hydrophilic domains of the simulated Nafion, a sulfonated tetrafluoroethylene polymeric system with hydration level 6 at 298 K. Adapted with permission from Reference [37] Copyright © 2011, American Chemical Society publishing.

**Figure 8 biomolecules-12-01473-f008:**
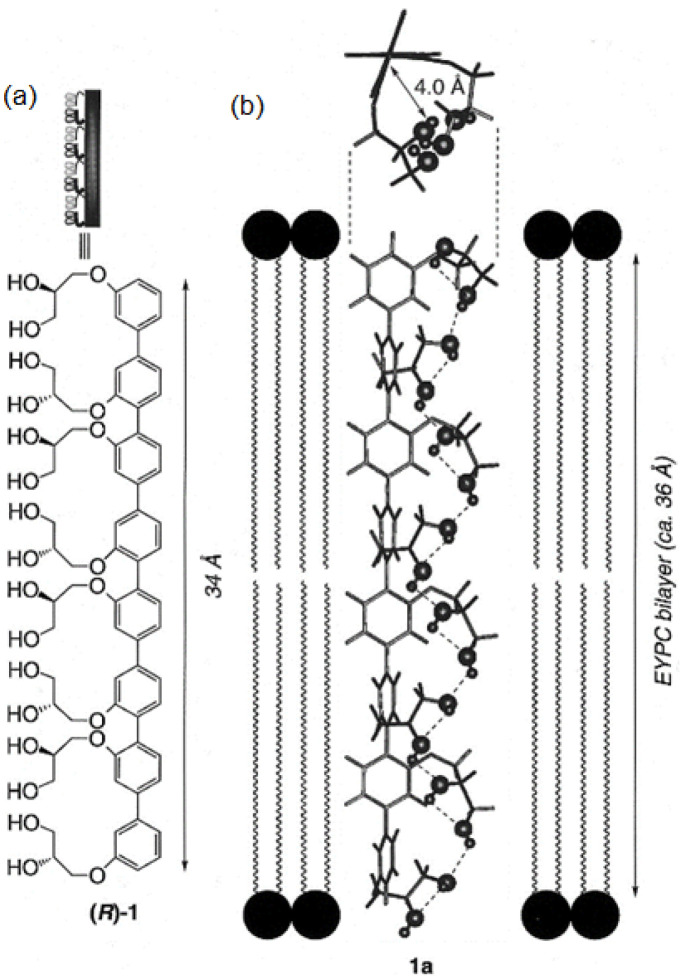
(**a**) The molecular structure of the rigid-rod polyol and (**b**) the 3D-drawing model using Molecular Simulations. The molecular model indicates the adopted conformation of the octameric polyol to be able to form the intramolecular hydrogen-bonded chains. Adapted with permission from Reference [46] Copyright © 1997, American Chemical Society publishing.

**Figure 9 biomolecules-12-01473-f009:**
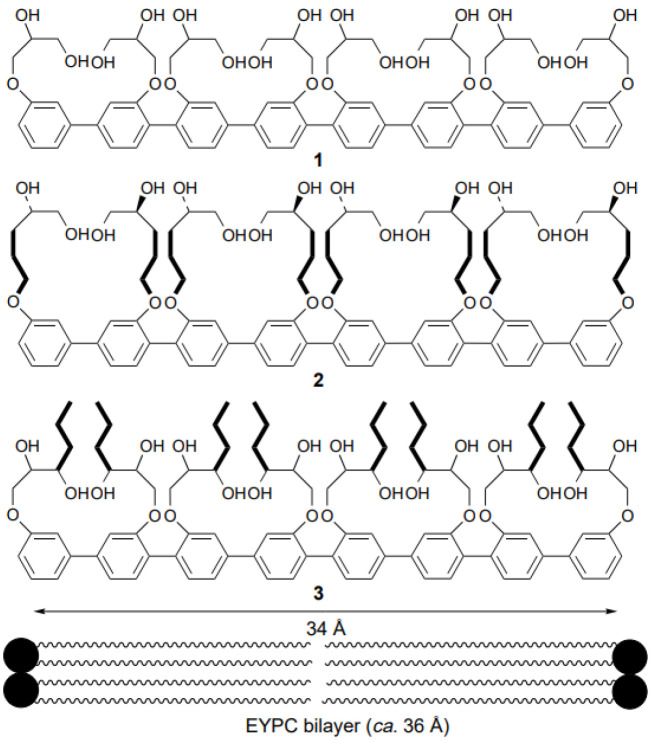
The chemical structures of the functionalized rigid-rod polyols tested for ion activity. Adapted with permission from Reference [47] Copyright © 1998, Royal Society of Chemistry publishing.

**Figure 10 biomolecules-12-01473-f010:**
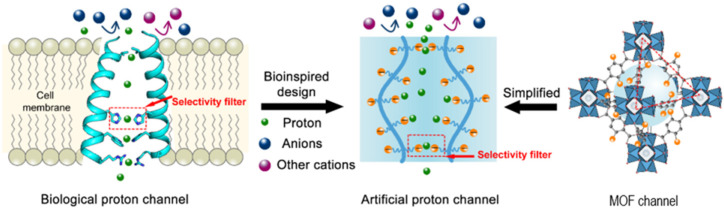
The schematic representation of the bioinspired selective proton MOF channels. Adapted with permission from Reference [50] Copyright © 2020, American Chemical Society publishing.

**Figure 11 biomolecules-12-01473-f011:**
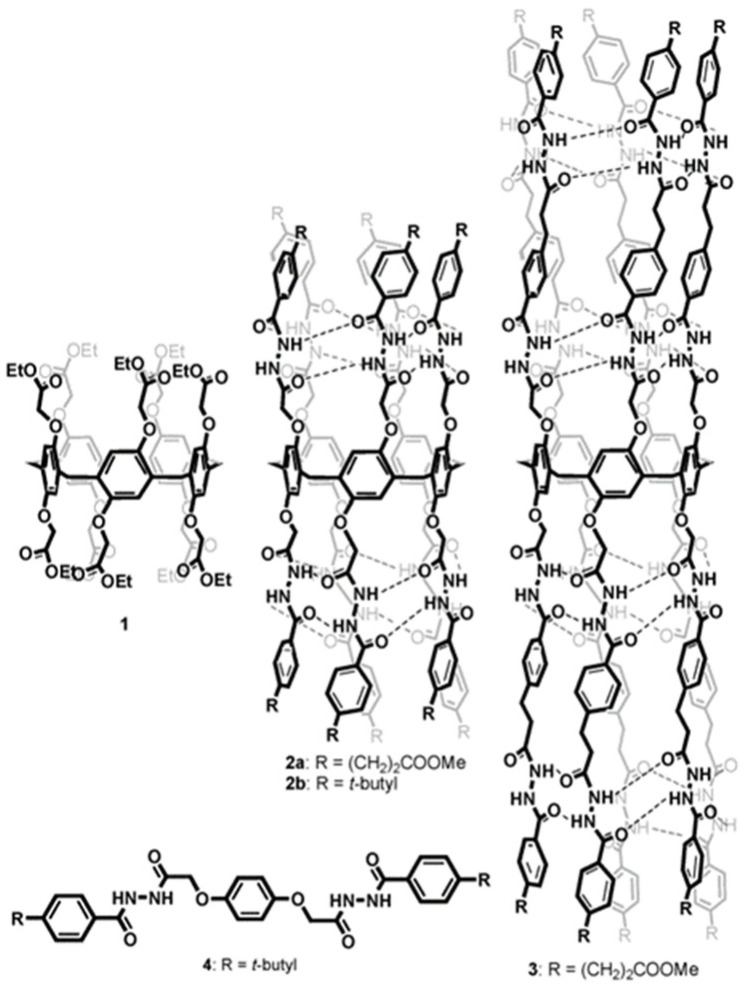
Self-assembled PAs tubular structure through intramolecular hydrogen bonding. Adapted with permission from Reference [53] Copyright © 2012, American Chemical Society publishing.

**Figure 12 biomolecules-12-01473-f012:**
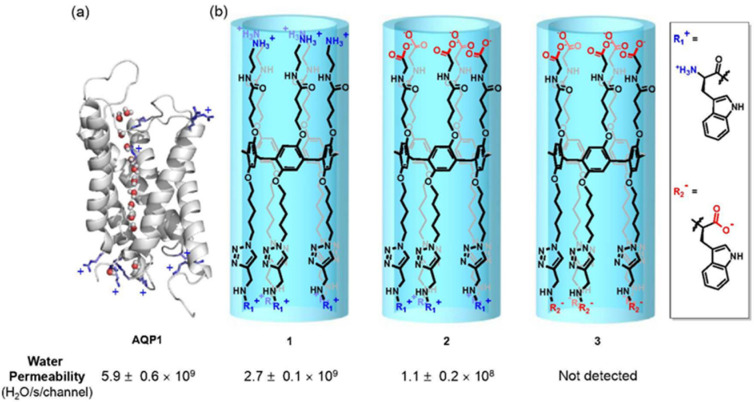
Structures and water permeability values for (**a**) aquaporin and (**b**) artificial pillar[5]arene channels prepared by attaching different side chains onto the PA backbone. Adapted with permission from Reference [54] Copyright © 2021, American Chemical Society publishing.

**Figure 13 biomolecules-12-01473-f013:**
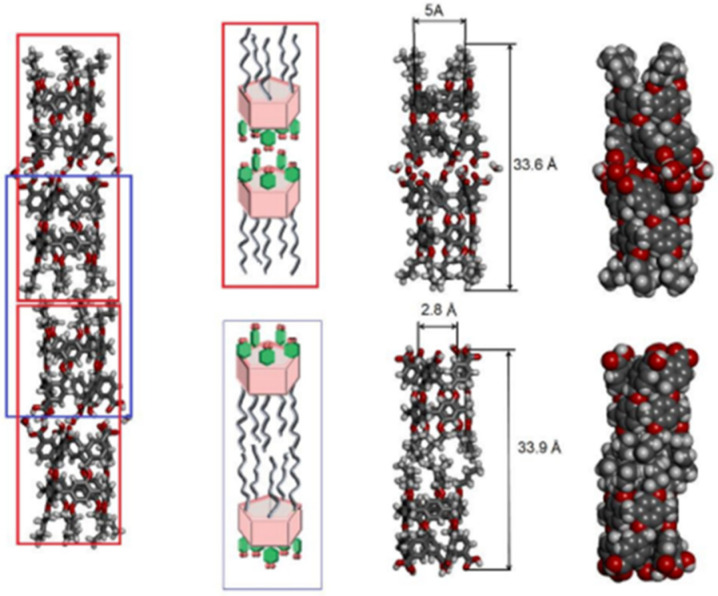
PA dimers structure obtained by Molecular Dynamics simulations with a total length of 34 Å. Adapted with permission from Reference [47] Copyright © 2020, Wiley publishing.

**Figure 14 biomolecules-12-01473-f014:**
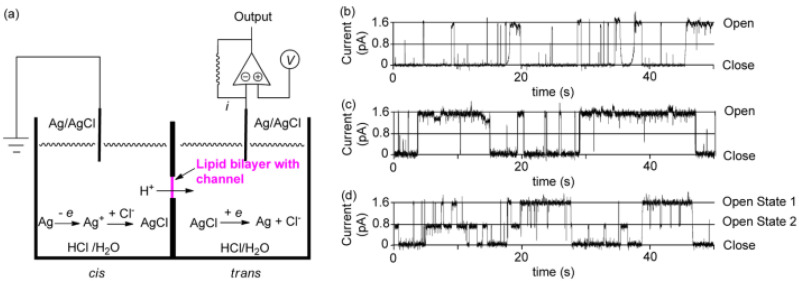
The representation of the patch-clamp experiments on lipid bilayer. (**a**) Schematic representation of the experimental set-up. (**b**) The current traces at +40 mV in planar lipid bilayer membranes in hydrochloric acid solution (pH 4.4). Adapted with permission from Reference [8] Copyright © 2011, Wiley publishing.

**Figure 15 biomolecules-12-01473-f015:**
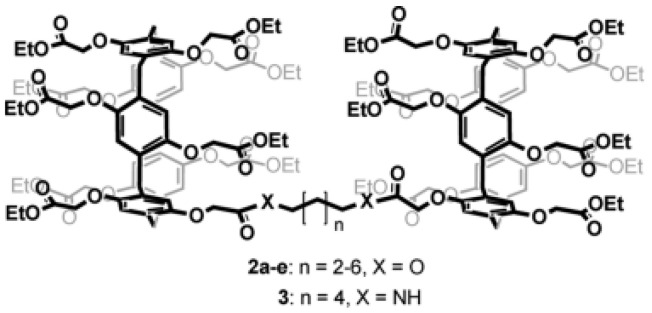
The chemical structural differences between the synthesized series of Pillar[5]arene derivatives. Adapted with permission from Reference [8] Copyright © 2011, Wiley publishing.

**Figure 16 biomolecules-12-01473-f016:**
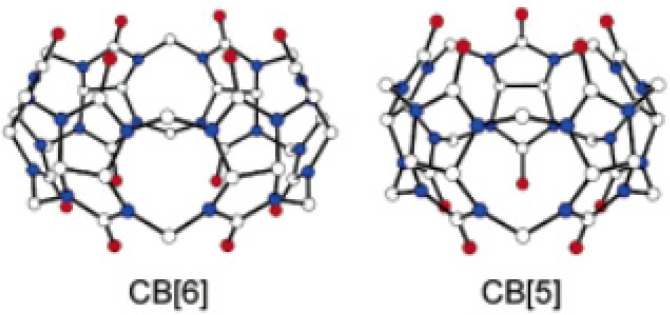
The structural representation of Cucurbit[n]uril derivatives. Adapted with permission from Reference [57] Copyright © 2004, American Chemical Society publishing.

**Figure 17 biomolecules-12-01473-f017:**
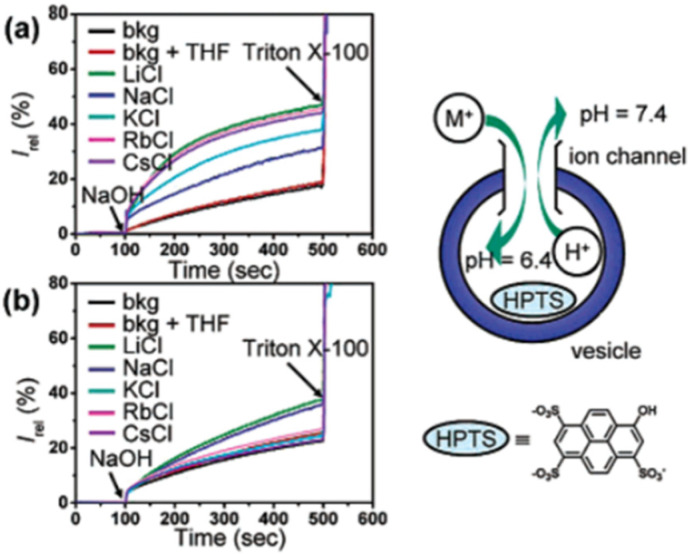
Changes in the fluorescence intensity ratio (I_460_/I_403_) as a function of time for EYPC vesicles: (**a**) in the presence of Cucurbit[6]uril; (**b**) in the presence of Cucurbit[5]uril. Adapted with permission from Reference [57] Copyright © 2004, American Chemical Society publishing.

**Figure 18 biomolecules-12-01473-f018:**
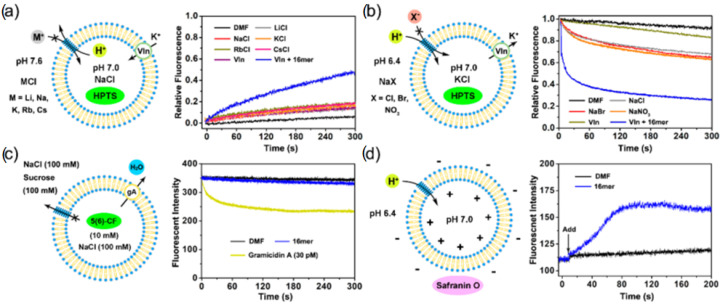
(**a**) pH-sensitive HPTS assay for assessing the cation transport (H^+^/M^+^ antiport- opposite directional transport of the species) activities of 16 mer by EYPC-based LUVs, buffered by HEPES with intravesicular NaCl (pH = 7.0, 100 mM) and extravesicular MCl (M = Li, Na, K, Rb, Cs, pH = 7.6, 100 mM), respectively. (**b**) pH-sensitive HPTS assay for assessing the anion transport (H^+^/X^−^ symport = unidirectional transport of the both species) activities of 16 mer by EYPC-based LUVs, buffered by HEPES with intravesicular KCl (pH = 7.0, 100 mM) and extravesicular NaX (X = Cl, Br^−^, NO_3_^−^, pH = 6.4, 100 mM), respectively. (**c**) Water transport activity of 16 mer (5 μM) and gA (30 pM) was measured by utilizing the self-quenching property of 5(6)-CF (10 mM, λex = 492 nm, λem = 517 nm) loaded inside the EYPC-based LUVs, buffered at 7.5 by HEPES with NaCl solution (100 mM). (**d**) Membrane polarization measured by the fluorescence intensity variation of Safranin O (200 nM; λex = 522 nm; λem = 581 nm) loaded outside of the EYPC-based LUVs in Mes buffer solution (pH 6.4). In (**a**–**d**), [total lipid] = 100 μM. Adapted with permission from Reference [58] Copyright © 2021, American Chemical Society publishing.

**Figure 19 biomolecules-12-01473-f019:**
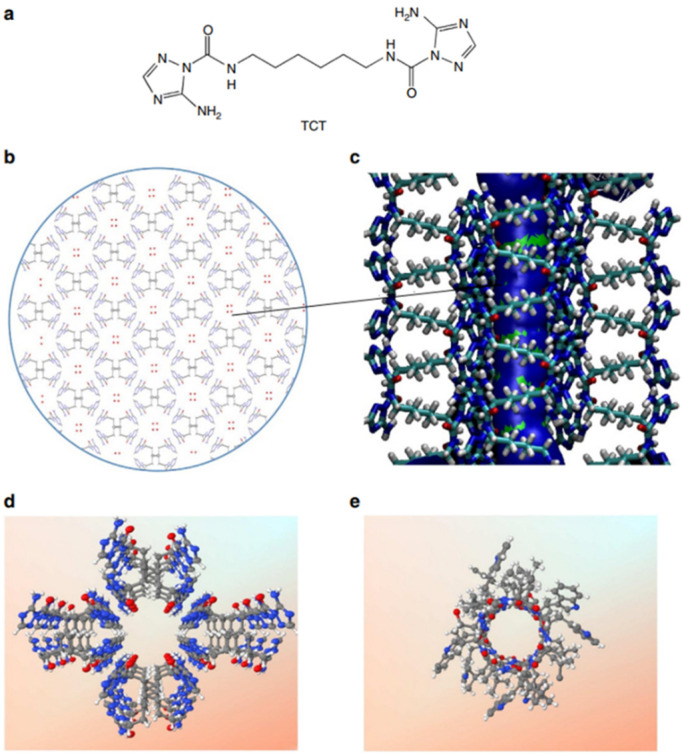
The molecular structure and the packing model of the TCT channel. (**a**) Chemical formula of TCT; (**b**) top and (**c**) side view of the T-channels in which water molecules (in (**b**)) are the red dots) pass through the pores; (**d**) is the T-channel; and (**e**) is the Gramicidin A channel. Adapted with permission from Reference [61] Copyright © 2014, Springer and Verlag publishing.

**Figure 20 biomolecules-12-01473-f020:**
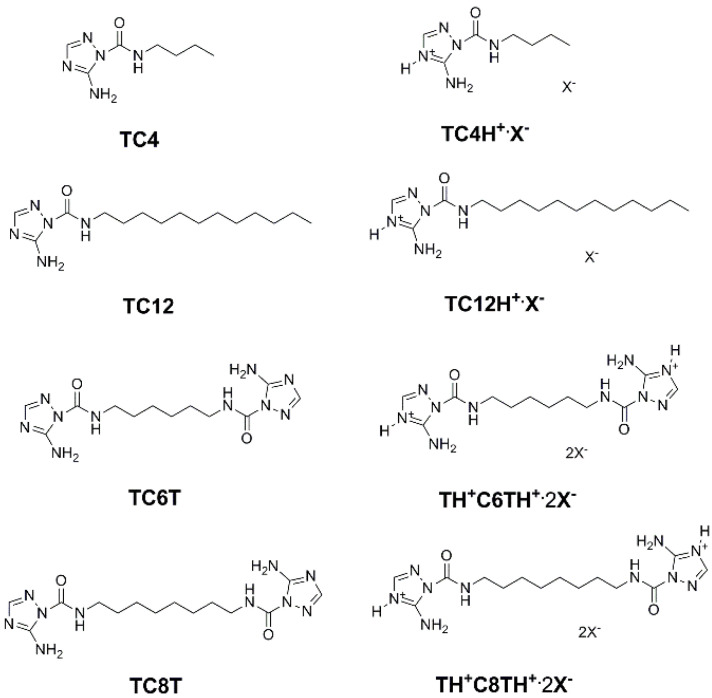
The molecular structure of the self-assembled amino-triazole amphiphiles and their protonated counterparts. Adapted with permission from Reference [63] Copyright © 2019, Wiley publishing.

**Figure 21 biomolecules-12-01473-f021:**
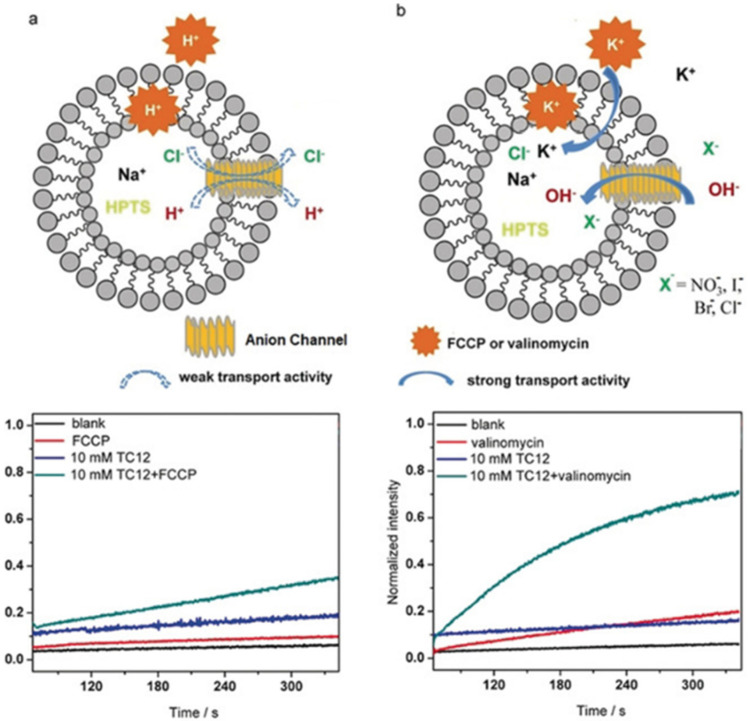
The proposed transport mechanism through the self-assembled columnar triazole channels and the comparison of the transport activity of one protonated molecule (**a**) in the absence and the presence of 50 μm FCCP (H^+^ carrier) and (**b**) in the absence and the presence of 50 μm valinomycin (K^+^ carrier). Adapted with permission from Reference [63] Copyright © 2019, Wiley publishing.

**Figure 22 biomolecules-12-01473-f022:**
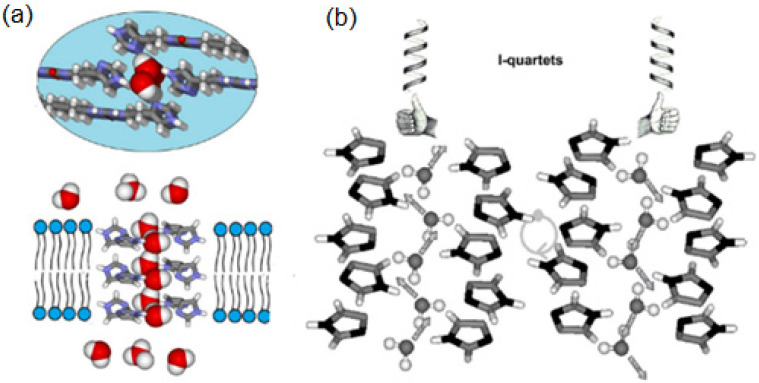
(**a**) Cross-section and top view of I-quartets channels generating water channels (**b**) enhancing the dipolar orientation of the water wires along the length of the channels. Adapted with permission from Reference [65] Copyright © 2011, Wiley publishing.

**Figure 23 biomolecules-12-01473-f023:**
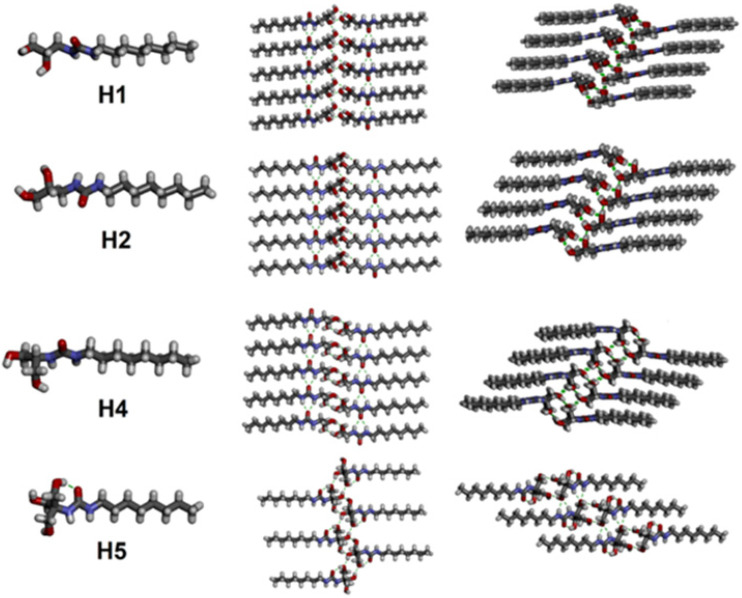
The single crystal structure of self-assembled OH channels. Adapted with permission from Reference [66] Copyright © 2021, American Chemical Society publishing.

**Figure 24 biomolecules-12-01473-f024:**
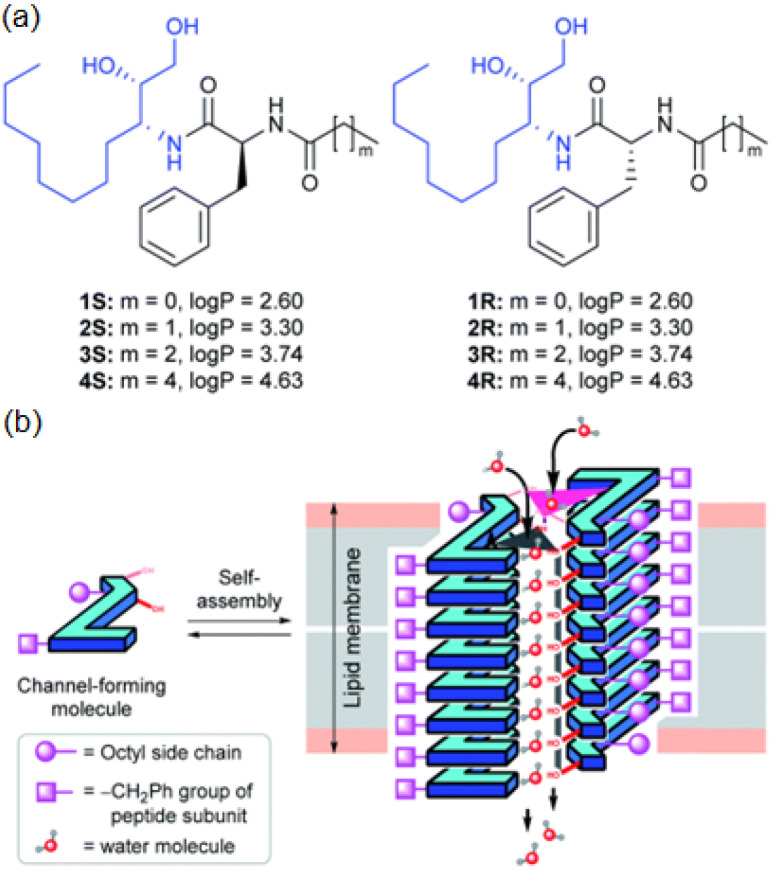
(**a**) The chemical structure of the peptide-diol isomers. (**b**) Schematic representation of the stacking dimer units self-assembled into water selective channel. Adapted with permission from Reference [69] Copyright © 2022, Royal Society Chemistry publishing.

**Figure 25 biomolecules-12-01473-f025:**
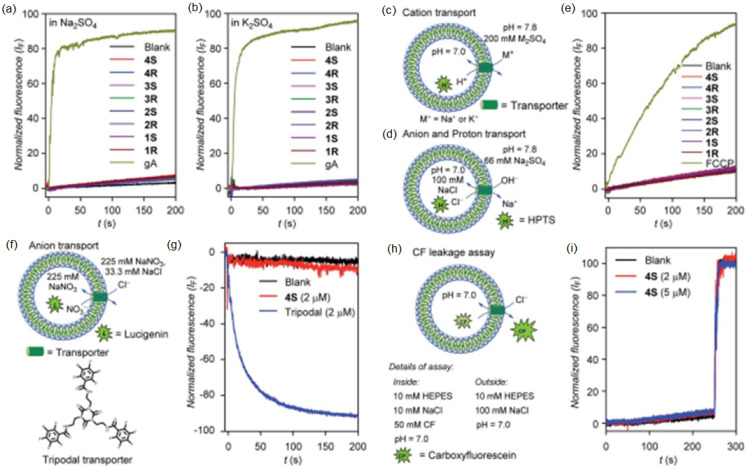
The Na^+^ (**a**) and K^+^ (**b**) transport activities using 2 μM compound in HPTS assays across EYPC vesicles. Cation (**c**) and anion (**d**) transport experiments. H^+^ transport activity of FCCP (0.2 μM) across EYPC vesicle (**e**). Vesicular representation (**f**) and anion transport activities (**g**) in lucigenin HPTS assays (2 μM) across EYPC vesicles (**i**). CF leakage assays (**h**). Adapted with permission from Reference [69] Copyright © 2022, Royal Society Chemistry publishing.

**Figure 26 biomolecules-12-01473-f026:**
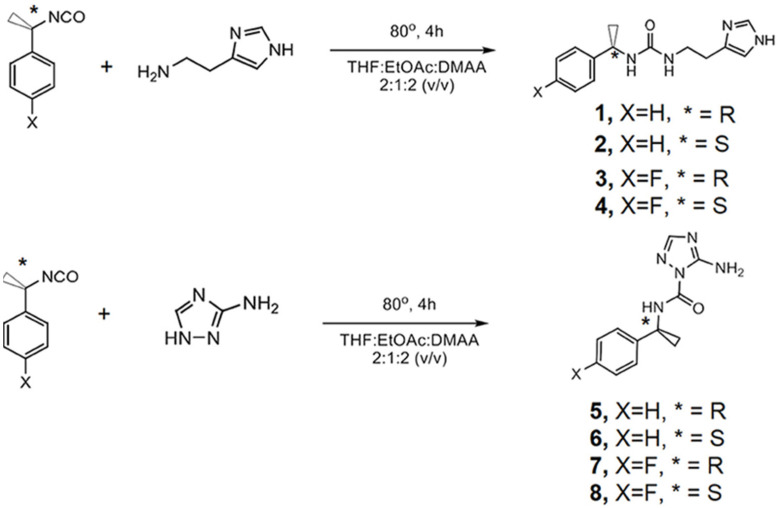
Structures of the synthesized chiral (*) fluoro-benzene-ureido imidazole or amino-triazole compounds involved in ion and proton transport across bilayer membranes. Adapted with permission from Reference [70] Copyright © 2021, Frontiers Media SA publishing.

**Figure 27 biomolecules-12-01473-f027:**
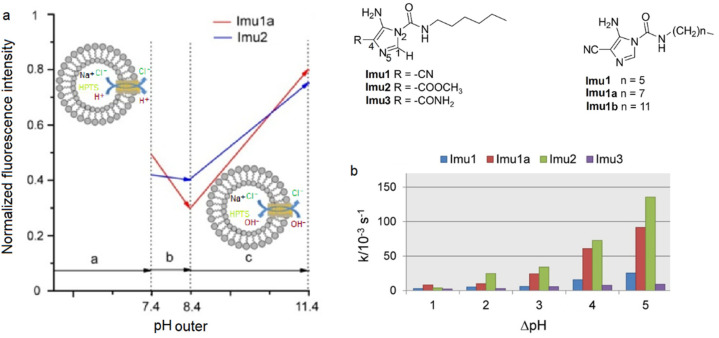
Structures of the synthesized amino-imidazole amphiphiles compounds. (**a**) Normalized fluorescence intensity vs. outer pH. (**b**) Pseudo-first-order initial transport rate constants vs. ΔpH between intra- and extravesicular media. Adapted with permission from Reference [71] Copyright © 2020, Wiley publishing.

**Figure 28 biomolecules-12-01473-f028:**
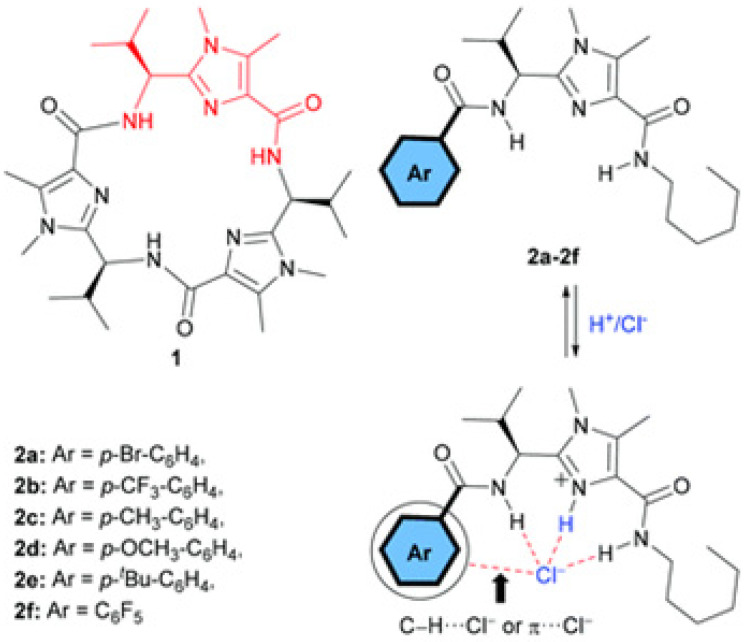
Structures of bis(amido)-imidazoles analogs of analog of marine cyclopeptide alkaloid westiellamide 1. Adapted with permission from Reference [72] Copyright © 2020, Royal Society of Chemistry publishing.

## Data Availability

Not applicable.

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
