# Peer review of "Biomimetic Artificial Proton Channels"

_biomolecules, 2022, doi:10.3390/biom12101473_

Round 1

Reviewer 1 Report

Please, read attached file.

Author Response

Many thanks for this positive report. The main choices were done by the first author of the paper and we tried to cover a majority of artifical systems used for the proton transport

Reviewer 2 Report

This review presents the current state of investigations aimed at designing and creating artificial proton channels. Since I am not an expert in this area, I take for granted that all the relevant work in the field has been referenced, and this work will serve as reference for all interested readers. I have, however, certain suggestions that could make this paper more intelligible for non-specialists.

1.       It would be useful to present, at the beginning, the existing models of intermolecular proton transfer. In particular, the Grotthuss mechanism, referenced several times in the manuscript, should be discussed in more detail. As of now, it is briefly mentioned on p.7, but in my opinion, the description should be extended (a figure?) and moved to the introduction.

2.       Several notions appear in the text which will sound unknown to many readers, e.g., “symport”, “antiport”. These should be explained, otherwise, this work will be understood only by people working in the field.

Minor comments:

-          l.120: O2: “2” should be written as subscript

-          l.151: of Hv1 are known: the context suggests the opposite

-          l.188: the hydrodynamics of water through the channels: water transfer?

-          l.278-9: performing excellent water-absorption performance

-          l.302-303: It was observed that their special behaviors…: something is missing in this sentence

-          l.597: along the a partly

-          l.606: Under symmetrical pH conditions: what does this mean?

-          l.667-668: have promising results…and understand

-          l.685: in regards of: in regard to/with respect to

Author Response

  1. It would be useful to present, at the beginning, the existing models of intermolecular proton transfer. In particular, the Grotthuss mechanism, referenced several times in the manuscript, should be discussed in more detail. As of now, it is briefly mentioned on p.7, but in my opinion, the description should be extended (a figure?) and moved to the introduction.

Authours reply: The ntermolecular proton transport is appearing several times in the manuscript. So it is better to keep it in each part and to keep our introduction as is it.

  1. Several notions appear in the text which will sound unknown to many readers, e.g., “symport”, “antiport”. These should be explained, otherwise, this work will be understood only by people working in the field.

Authors reply : done

Minor comments:

  •          l.120: O2: “2” should be written as subscript- Done

-          l.151: of Hv1 are known: the context suggests the opposite- we removed the conflictual context

-          l.188: the hydrodynamics of water through the channels: water transfer? -yes

-          l.278-9: performing excellent water-absorption performance -replaced with water swelling

-          l.302-303: It was observed that their special behaviors…: something is missing in this sentence - we corrected this sentence

-          l.597: along the a partly corrected

  •          l.606: Under symmetrical pH conditions: what does this mean? corrected with  : ....using the same pH in the feed and receiving phases 
  •          l.667-668: have promising results…and understand: replaced with : ....have been shown promising results on efficient transport like in natural systems,

-          l.685: in regards of: in regard to/with respect to. ok corrected

Reviewer 3 Report

The manuscript by Andrei and Barboiu reviews recent contributions on the topic of biomimetic artificial proton channels. This topic is very relevant and interesting for the audience of MDPI’s Biomolecules. The number of references covered in this review is sufficient to justify the manuscript, while not being too overwhelming. There are no major concerns to the manuscript, however, I’d like to pinpoint two issues.

In the abstract, some words are combined in a sentence without proper correlation. For instance, when it’s stated that “details” are “inconclusive” (line 9), or that the “understanding” of the phenomenon is still “undecided” (lines 11 and 12). How can a detail be inconclusive, itself? Research or evidence can be inconclusive. And how can an understanding be undecided? It may be disputed among researchers, for sure. I believe these tiny issues can be corrected during English editing/reviewing of the manuscript. Moreover, the abstract could be reformulated to better describe what is being addressed in the manuscript, as lines 7 to 14 are devoted just to justifying the importance of the matter in what may feel like a non-quantifiable and opinionated way. Actually, the whole manuscript has many opinionated sentences, e.g., when the “unique and quite important” properties of Hv1 channels are commented, on line 138. Although this style of writing is not usual for scientific texts, I can understand how it may seem fit for a review. Nevertheless, I strongly suggest searching the manuscript to remove the excess of opinionated sentences.

Finally, the use of an English editing/reviewing service is highly recommended to correct minor typos and for adequate spellchecking. Below I’m citing some needed corrections, although most of them I did not annotate:

-       line 83, correct the subscript and superscript characters for the bicarbonate anion;

-       line 116, change “were” for “was”;

-       line 120, correct the subscript for the oxygen molecule;

-       line 134, correct the superscript for the proton positive charge;

-       line 160, change “swoed” for “showed”;

-       line 257, change “form successfully” for “successfully form”;

-       line 323, change “threatening” for “treating”;

-       line 476, change “internally” for “internal”;

-       line 563, remove the end stop before “(Figure 25)”;

-       line 678, change “insight of” for “insight on”;

-       and many other small issues that should be checked for.

---- 

Author Response

The manuscript by Andrei and Barboiu reviews recent contributions on the topic of biomimetic artificial proton channels. This topic is very relevant and interesting for the audience of MDPI’s Biomolecules. The number of references covered in this review is sufficient to justify the manuscript, while not being too overwhelming. There are no major concerns to the manuscript, however, I’d like to pinpoint two issues.

In the abstract, some words are combined in a sentence without proper correlation. For instance, when it’s stated that “details” are “inconclusive” (line 9), or that the “understanding” of the phenomenon is still “undecided” (lines 11 and 12). How can a detail be inconclusive, itself? Research or evidence can be inconclusive. And how can an understanding be undecided?

Authors reply: We corrected these sentences in order to avoid such confusion

It may be disputed among researchers, for sure. I believe these tiny issues can be corrected during English editing/reviewing of the manuscript. Moreover, the abstract could be reformulated to better describe what is being addressed in the manuscript, as lines 7 to 14 are devoted just to justifying the importance of the matter in what may feel like a non-quantifiable and opinionated way. Actually, the whole manuscript has many opinionated sentences, e.g., when the “unique and quite important” properties of Hv1 channels are commented, on line 138. Although this style of writing is not usual for scientific texts, I can understand how it may seem fit for a review. Nevertheless, I strongly suggest searching the manuscript to remove the excess of opinionated sentences.

Authors repliy: We removed the opinonated sentences

Finally, the use of an English editing/reviewing service is highly recommended to correct minor typos and for adequate spellchecking. Below I’m citing some needed corrections, although most of them I did not annotate:

-       line 83, correct the subscript and superscript characters for the bicarbonate anion; corrected

  •       line 116, change “were” for “was”; corrected

-       line 120, correct the subscript for the oxygen molecule; corrected

-       line 134, correct the superscript for the proton positive charge; corrected

-       line 160, change “swoed” for “showed”; corrected

-       line 257, change “form successfully” for “successfully form”; corrected

-       line 323, change “threatening” for “treating”; corrected

-       line 476, change “internally” for “internal”;  corrected

-       line 563, remove the end stop before “(Figure 25)”; done

-       line 678, change “insight of” for “insight on”; corrected

-       and many other small issues that should be checked for. we have checked carrefuly the manuscript and get the corrections